# Associations of Vitamin D Deficiency, Parathyroid hormone, Calcium, and Phosphorus with Perinatal Adverse Outcomes. A Prospective Cohort Study

**DOI:** 10.3390/nu12113279

**Published:** 2020-10-26

**Authors:** Íñigo María Pérez-Castillo, Tania Rivero-Blanco, Ximena Alejandra León-Ríos, Manuela Expósito-Ruiz, María Setefilla López-Criado, María José Aguilar-Cordero

**Affiliations:** 1Andalusian Plan for Research, Development and Innovation, CTS 367, University of Granada, 18001 Granada, Spain; perezcastillo@correo.ugr.es (Í.M.P.-C.); taniarivero89@gmail.com (T.R.-B.); ximenaleonr18@gmail.com (X.A.L.-R.); 2Foundation for Biomedical Research in Eastern Andalusia (FIBAO), 18014 Granada, Spain; manuela.exposito.ruiz@juntadeandalucia.es; 3Obstetrics and Gynecology Service, Virgen de las Nieves University Hospital, 18014 Granada, Spain; mmefilla@gmail.com; 4Department of Nursing, Faculty of Health Sciences, University of Granada, 18071 Granada, Spain

**Keywords:** vitamin D deficiency, perinatal adverse outcomes, 25-hydroxyvitamin D, parathyroid hormone, PTH, calcium, phosphorus, cohort study

## Abstract

Vitamin D deficiency during pregnancy has been linked to perinatal adverse outcomes. Studies conducted to date have recommended assessing interactions with other vitamin D-related metabolites to clarify this subject. We aimed to evaluate the association of vitamin D deficiency during early pregnancy with preterm birth. Secondary outcomes included low birth weight and small for gestational age. Additionally, we explored the role that parathyroid hormone, calcium and phosphorus could play in the associations. We conducted a prospective cohort study comprising 289 pregnant women in a hospital in Granada, Spain. Participants were followed-up from weeks 10–12 of gestation to postpartum. Serum 25-hydroxyvitamin D, parathyroid hormone, calcium, and phosphorus were measured within the first week after recruitment. Pearson’s χ^2^ test, Mann–Whitney U test, binary and multivariable logistic regression models were used to explore associations between variables and outcomes. 36.3% of the participants were vitamin D deficient (<20 ng/mL). 25-hydroxyvitamin D concentration was inversely correlated with parathyroid hormone (ρ = −0.146, *p* = 0.013). Preterm birth was associated with vitamin D deficiency in the multivariable model, being this association stronger amongst women with parathyroid hormone serum levels above the 80th percentile (adjusted odds ratio (aOR) = 6.587, 95% CI (2.049, 21.176), *p* = 0.002). Calcium and phosphorus were not associated with any studied outcome. Combined measurement of 25-hydroxyvitamin D and parathyroid hormone could be a better estimator of preterm birth than vitamin D in isolation.

## 1. Introduction

Vitamin D deficiency is considered to be a pandemic [1] whose global prevalence varies widely depending on the studied population, dietary intake, ultraviolet-B light exposure, ethnicity, and age, amongst other factors [2]. The severe deficiency of this secosteroid is associated with skeletal disorders as well as other pathologies outside bone metabolism [3]. During pregnancy, vitamin D deficiency has been linked to pregnancy and perinatal adverse outcomes such as pre-eclampsia, gestational diabetes mellitus, preterm birth, and low birth weight [4].

Preterm birth (PTB) is the leading cause of mortality in children under five years old worldwide, and its global prevalence has been estimated to be 10.6% of all births accounting for 14.84 million newborns in 2014 [5]. PTB is regarded as a syndrome resulting from different mechanisms such as uteroplacental dysfunction, inflammation, and infection, along with other immunological processes [6]. Vitamin D exerts important immunomodulatory effects decreasing levels of IL-1, IL-6 and TNF-α produced by macrophages [7], regulating the activity of lymphocytes B and T [8], and inducing human cathelicidin production [9], thus playing an important role in both innate and adaptive immune responses. However, studies evaluating the association between vitamin D deficiency during pregnancy and prematurity have not reached consensus on their results [10,11].

According to the WHO, low birth weight (LBW) is defined as a birth weight of less than 2500 g [12], whereas small for gestational age (SGA) is defined as weight below 10th percentile for the gestational age and depends on the reference population [13]. Several authors have associated vitamin D deficiency during pregnancy with LBW and SGA [14,15,16,17]. Possible mechanisms of action of vitamin D on fetal growth might consist of anti-inflammatory properties, regulation of genes implicated in angiogenesis, promotion of trophoblast invasion and control of fetal glucose availability [13,14,16,18]. Meta-analyses conducted to evaluate the associations between vitamin D deficiency, PTB, LBW or SGA have not yielded strong evidence [19,20,21,22,23].

25-hydroxyvitamin D (which will be referred to as vitamin D throughout the paper) plays a key role in calcium and phosphorus homeostasis and its concentration is regulated by the parathyroid hormone (PTH). However, vitamin D is usually measured in isolation and some authors have highlighted the importance of assessing interactions with vitamin D-related metabolites when evaluating associations with pregnancy and perinatal outcomes [24,25,26]. In this regard, low maternal calcium concentrations have been associated with LBW [27] and PTB [17,28], but evidence remains unclear. Furthermore, several authors have described the concept of functional vitamin D deficiency characterized by secondary hyperparathyroidism, which refers to elevated levels of PTH in combination with low levels of vitamin D [26,29]. According to this concept, calcium metabolic stress rather than vitamin D insufficiency would be an etiological factor for fetal growth impairment as a consequence of secondary hyperparathyroidism. In line with this idea, a recent study has suggested that the combined measurement of 25-hydroxyvitamin D and PTH during pregnancy could be a better determinant of fetal growth restriction [25]. In this same study, calcium levels were elevated only among pregnant women with high PTH levels in combination with low concentrations of vitamin D [25]. Therefore, calcium measurement proved to be an interesting addition to previous studies [26,29]. Finally, the synthesis of 1,25-dihydroxyvitamin D, which is the most active form of vitamin D, is closely regulated by PTH, calcium and phosphorus [3]. Hence, 25-hydroxyvitamin D deficiency could not be indicative of low levels of 1,25-dihydroxyvitamin D without considering the impact that other metabolites might have on the association [24].

The main purpose of the present research is to study the association between vitamin D deficiency during early pregnancy and PTB. Secondary outcomes of the study consist of evaluating the influence of vitamin D deficiency during pregnancy on the odds of LBW, and SGA, as well as to explore the role that metabolites related to the metabolism of this secosteroid, namely PTH, calcium, and phosphorus, could play in the associations. We hypothesized that vitamin D deficiency defined as maternal 25-hydroxyvitamin D levels below 20 ng/mL is associated with higher odds of preterm birth, low birth weight and small for gestational age, these associations being stronger among women with parathyroid hormone levels above the 80th percentile.

## 2. Materials and Methods

To achieve the proposed objectives, we conducted a prospective cohort study at the University Hospital Complex “Virgen de las Nieves” of Granada, Spain, a medical center with 2956 deliveries in 2019. Pregnant women were recruited from 2018 to 2019 and followed-up from weeks 10–12 of gestation to one month postpartum. This study was approved by the Ethics Committee of the University of Granada, number 72-2015, and conducted in accordance with the principles of the Declaration of Helsinki, reviewed in Fortaleza, Brazil, in 2003. Results of the present study are reported following the STROBE statement guidelines for cohort studies [30].

### 2.1. Participants Data

Women were approached in their first prenatal visit at the obstetrics and gynecology services of the hospital complex. Inclusion criteria included pregnant women older than 16 years old, able to speak Spanish, and capable of signing for informed consent between 10–12 weeks of gestation determined by ultrasonography. Exclusion criteria at enrollment consisted of pregnant women with the intention to give birth in a different hospital. Other exclusion criteria consisted of women undergoing voluntary interruption of pregnancy, miscarriage, stillbirth, and multiple pregnancy. Previous history of pregnancy adverse outcomes was not an exclusion criterion for the present study.

The required sample size for the present study was calculated based on the results obtained in another study conducted by Perez-Ferre et al., who observed a prevalence of preterm birth amongst vitamin D deficient women (<20 ng/mL) of 22.9% and a prevalence of preterm birth amongst vitamin D sufficient women (>20 ng/mL) of 8.25% with a vitamin D sufficiency/deficiency ratio of 0.69 [31]. To achieve a power of 80% to detect differences in the null hypothesis H_0_:p1 = p2, using χ^2^ test with a confidence level of 95%, we estimated a sample size of 203 participants. Given the prospective design of the study, we estimated 20% of lost to follow-up. Hence, final calculated minimum sample size consisted of 244 participants to be included in the study.

Sociodemographic characteristics of participants were collected at recruitment by researchers from self-report and medical records. Considered variables consisted of maternal age, pre-gestational body mass index (BMI), smoking habit during pregnancy (defined as >1 or 0 cigarettes per day), parity and gravidity, history of previous pregnancy, and perinatal adverse outcomes (LBW, SGA, PTB, pre-eclampsia, gestational diabetes mellitus, miscarriage, and stillbirth), ethnicity and seasonality of sampling. Women with pre-gestational BMI >30 were classified as obese. Data regarding vitamin D supplementation at recruitment was not collected. However, Spain is a country without vitamin D supplementation policy, and vitamin D supplementation among Spanish pregnant women is uncommon in comparison with other European countries [32].

### 2.2. Clinical and Biochemical Procedures

Fasting maternal blood samples were obtained during the week of enrolment. Sampling was performed by venipuncture in tubes containing anticoagulant (EDTA, Ethylenediaminetetraacetic acid) and were immediately transported to the laboratory for analysis.

25-hydroxyvitamin D and intact-PTH (1–84) were quantified by microparticle chemiluminescence immunoassay (CMIA) using an Alinity I^®^ analyzer (Abbott, Wiesbaden, Germany). Briefly, CMIA analysis is based on the use of paramagnetic microparticles coated with antibodies. Regarding 25-hydroxyvitamin D, it is first separated from the vitamin D-binding protein (DBP) to be mixed with the anti-vitamin D antibody-coated microparticle. The complex is labeled with acridinium afterwards. The reaction conjugate is incubated to be later washed-out, and the correlation between emitted chemiluminescence light measured in relative light units (RLU) and the 25-hydroxyvitamin D or intact-PTH concentration is calculated. According to the manufacturer, the method detection limit for the 25-hydroxyvitamin D assay is 3.5 ng/mL (8.85 nmol/L) and intra-assay coefficient of variation is 3.6% at 39.8 ng/mL (99.4 nmol/L) whilst the quoted PTH assay detection limit is 0.5 pg/mL (0.05 pmol) and intra-assay coefficient of variation is 2.6% at 63.8 pg/mL (6.76 pmol/L).

Calcium and phosphorus were analyzed using an Alinity C^®^ analyzer (Abbott, Wiesbaden, Germany). Calcium was analyzed by arsenazo-III colorimetric assay measuring absorbance at 660 nm whilst phosphorus was analyzed by phosphomolybdate assay measuring absorbance at 340 nm.

Vitamin D deficiency was defined as serum 25-hydroxyvitamin D concentrations <20 ng/mL (50 nmol/L) whilst vitamin D insufficiency was defined as serum 25-hydroxyvitamin D concentrations <30 ng/mL (75 nmol/L). Used cut-off points were based on other studies [22]. Chosen cut-off points differed from those recommended by the American Institute of Medicine [33], however, optimal vitamin D cut-off points during pregnancy remain controversial and consensus on this matter has not been reached to date [34]. The seasonality of sampling was considered as a potential confounder given the existing association between sun exposure and vitamin D concentration [35]. Due to a lack of consensus, we considered elevated PTH levels as concentrations above the 80th percentile in line with another author [26]. Therefore, women with elevated PTH levels were those with PTH concentrations ≥31.9 pg/mL.

Women were followed-up in subsequent prenatal visits, and cases of pre-eclampsia and gestational diabetes mellitus were diagnosed. Values of maternal diastolic and systolic blood pressure, proteinuria, and glucose tolerance test results were collected by researchers during routine controls. Blood pressure was measured using a validated automatic tensiometer and the measurement was repeated within 15 min. De novo systolic blood pressure >140 mm/Hg and diastolic blood pressure >90 mm/Hg measurements were considered as gestational hypertension and women were further evaluated by the obstetrician. Proteinuria was defined as urine protein-to-creatinine ratio above 0.3 mg/mg and was assessed in routine controls after week 20 of gestation. Proteins in urine were quantified using benzethonium chloride turbidimetric method and creatinine was analyzed using alkaline picrate colorimetric assay. An oral glucose tolerance test was performed between weeks 24–28 of gestation. Blood glucose was analyzed using a hexokinase/glyceraldehyde 3-phosphate dehydrogenase activity assay kit.

Cases of pre-eclampsia were defined according to the International Society for the Study of Hypertension in Pregnancy (ISSHP) 2018 classification [36], and GDM cases were defined in line with the American Diabetes Association criteria [37]. Cases of miscarriage and stillbirth, type of delivery, and values of gestational age at delivery and birth weight were documented from medical records. Low birth weight was defined as live birth with less than 2500 g at delivery in accordance with the International Classification of Diseases, 10th Edition [12]. Preterm birth was defined as live birth with less than 37 weeks of gestation [38]. Small for gestational age cases were considered as live births with weight below 10th percentile for the gestational age [13] and were calculated using Spanish reference percentile charts from 2010–2014, based on gender, parity, and type of delivery [39].

### 2.3. Statistical Analysis

All statistical analyses were performed using the software SPSS version 25 (IBM Corp^®^, Armonk, NY, USA). Normality of continuous variables was examined using Kolmogorov–Smirnov test. Categorical variables were reported as percentages, and continuous variables were reported as mean ± standard deviation or median and interquartile range based on normality test results. Differences between participants depending on vitamin D cut-off points were analyzed using Pearson’s chi-square (χ^2^) test for categorical variables and the Mann–Whitney U test for continuous variables. The Spearman correlation test was used to evaluate the strength of association between vitamin D and concentrations of PTH, calcium, and phosphorus. A scatter plot was provided to graphically represent statistically significant correlations. For each outcome, bivariate analysis was performed to evaluate possible confounders based on the literature. Variables with *p*-values < 0.20 in bivariate analysis were chosen for adjustment in multivariable analysis. This cut-off is supported by the literature [40,41]. Other related variables strongly supported by the scientific literature were also considered for adjustment when applicable. Odds ratios (ORs) with 95% confidence interval (95% CI) were calculated for each chosen outcome and biomarker using bivariate and multivariable logistic regression models. In logistic regression models, parathyroid hormone, calcium and phosphorus were analyzed as continuous variables whilst vitamin D deficiency was a categorical variable (<20 ng/mL/≥20 ng/mL)

Finally, we provided binary logistic regression unadjusted and adjusted models to examine the associations between concentrations of vitamin D <20 ng/mL and <30 ng/mL along with the PTH 80th percentile and the odds of PTB, LBW, SGA in the cohort of study. A sensitivity analysis was conducted to evaluate consistency of the results using the PTH 75th percentile.

## 3. Results

### 3.1. Cohort of Study

We approached 500 women for study participation, 380 of whom signed informed consent and were enrolled in the study. After follow-up, a completed dataset from 303 women and their children was available (20.26% lost to follow-up). A final analytical sample of 289 women fulfilled inclusion criteria and was available for the present study Figure 1.

### 3.2. Characteristics of Participants

The sociodemographic characteristics of participants based on vitamin D cut-off points (<20 ng/mL or ≥20 ng/mL), are presented in Table 1 and concentrations of calcium, phosphorus and parathyroid hormone are reported in Table 2. Results of the Kolmogorov–Smirnov test showed that maternal age, BMI, calcium, phosphorus, and PTH concentrations were non-normally distributed across vitamin D cut-off points. All expected numbers were higher than five in Pearson’s χ^2^ test for categorical variables. Vitamin D levels were normally distributed amongst participants. Serum 25-hydroxyvitamin D mean concentration was 22.36 ± 6.3 ng/mL. Thirty-four women had sufficient levels of vitamin D (≥30 ng/mL) (11.76%), 150 were vitamin D insufficient (20–29.9 ng/mL) (51.9%), and the 105 remaining women suffered vitamin D deficiency (<20 ng/mL) (36.33%). Median maternal age was 33 (29–36) years old, whilst the median pre-pregnancy BMI was 25.1 (21.9–29.3). 52 participants were obese (18%). With respect to the history of previous pregnancy adverse outcomes, 67 women had history of miscarriage or stillbirth; one had history of pre-eclampsia; five had history of gestational diabetes mellitus; and 12 women had history of preterm birth. Regarding ethnicity, three African women were lost to follow-up, and most of the ethnic women approached did not fulfill the inclusion criteria (speak Spanish). Therefore, all women who completed the study were Caucasian. Only obesity (pre-pregnancy BMI ≥30), preterm birth and maternal blood parathyroid hormone concentration varied significantly across the chosen vitamin D cut-off points (*p* < 0.05).

The spearman correlation test showed an inverse association between vitamin D and parathyroid hormone concentrations (ρ = −0.146, *p* = 0.013). This correlation was also evident in the scatter plot in Figure 2. On the other hand, neither calcium nor phosphorus were correlated with vitamin D in the Spearman’s test (calcium: ρ = 0.022, *p* = 0.705, phosphorus: ρ = −0.024, *p* = 0.689).

### 3.3. Pregnancy and Perinatal Adverse Outcomes

Frequencies of pregnancy and perinatal adverse outcomes observed in the present study compared to estimated global frequencies and estimated frequencies in the USA and Europe are described in Table 3. One pre-eclampsia case was a twin pregnancy, thus being excluded from further analyses. We also excluded type I and pre-gestational type II diabetes cases (four cases) when describing the frequency of gestational diabetes mellitus in the cohort of study.

With the exemption of LBW, frequencies of adverse outcomes in the cohort of study were lower than average estimated frequencies. Seventeen births were premature (<37 weeks of gestation) (5.9%) and 24 newborns had low birth weight (<2.500 g) (8.3%). When comparing gestational age and birth weight data with the Spanish reference percentile charts [39], we obtained a total of 27 SGA cases in the cohort of study (birth weight < 10th percentile for their gestational age) (9.34%).

### 3.4. Associations of Vitamin D Deficiency, PTH, Calcium, and Phosphorus with Perinatal Adverse Outcomes

In Table 4, unadjusted and adjusted logistic regression models are presented to describe associations between vitamin D deficiency (<20 ng/mL/<50 nmol/L), parathyroid hormone, calcium, phosphorus continuous concentrations, and perinatal outcomes. Covariables with *p*-values < 0.20 in bivariate analysis were selected for adjustment in multivariable analysis.

Maternal first-trimester vitamin D deficiency was associated with higher odds of preterm birth in bivariate analysis, but it was not statistically significant (OR = 2.662, 95% CI (0.982, 7.217), *p* = 0.054). Only after adjusting for history of PTB and cases of pre-eclampsia, did the association become statistically significant (OR = 3.529, 95% CI (1.159, 10.741), *p* = 0.026). PTH concentration and preterm birth were weakly associated only in bivariate analysis (OR = 1.030, 95% CI (1.002, 1.058), *p* = 0.035). Regarding birth weight, there was a trend towards higher odds of low birth weight amongst the offspring of vitamin D deficient women. However, this association was not statistically significant neither in bivariate analysis or after adjusting for confounders (OR = 2.222, 95% CI (0.958, 5.157), *p* = 0.06/aOR = 1.586, 95% CI (0.586, 4.336), *p* = 0.361). In the same fashion, the relationship between vitamin D deficiency and risk of SGA was not significant neither in crude or adjusted models (OR = 2.024, 95% CI (0.912–4.488), *p* = 0.083/aOR = 1.794, 95% CI (0.786–4.093), *p* = 0.165). We did not observe any correlation between calcium and phosphorus concentrations with perinatal outcomes.

In Table 5, we presented the associations between vitamin D deficiency and insufficiency along with the PTH 80th percentile and perinatal outcomes.

Overall, first-trimester vitamin D insufficiency defined as maternal blood levels of 25-hydroxyvitamin D < 30 ng/mL along with levels of PTH above the 80th percentile correlated with prematurity, but the association was not statistically significant (OR = 2.611, 95% CI (0.92, 7.411), *p* = 0.071). However, vitamin D deficiency (<20 ng/mL) during the first trimester of gestation was strongly associated with the odds of PTB amongst women with PTH levels above the 80th percentile (OR = 5.389, 95% CI (1.837, 15.812), *p* = 0.002). Furthermore, this association remained evident after adjusting for preterm birth confounders (aOR = 6.587, 95% CI (2.049, 21.176), *p* = 0.002). Vitamin D concentrations ≥ 20 ng/mL and PTH levels ≤ 80th percentile did not correlate with PTB (*p* > 0.05).

Low birth weight was more prevalent amongst women with vitamin D levels <20 ng/mL in combination with PTH levels > 80th percentile (OR = 4.135, 95% CI (1.560, 10.963), *p* = 0.004). However, this association was rendered statistically non-significant after adjusting for confounders (aOR = 2.653, 95% CI (0.766, 9.188), *p* = 0.124).

Finally, we did not find any association between SGA and vitamin D deficiency or insufficiency along with the 80th PTH percentile neither in crude nor adjusted models. Sensitivity analyses using the 75th PTH percentile (≥29.25 pg/mL) were performed to evaluate the consistency of the results Appendix A. Overall, associations between studied outcomes and combinations of vitamin D deficiency/insufficiency with the 75th PTH percentile were similar to those shown in the main analysis.

## 4. Discussion

The literature about deficiency of vitamin D and perinatal outcomes is inconsistent, and several authors have suggested that interactions with metabolites linked to the metabolism of vitamin D could play an important role in the associations [24,25,26]. We conducted a prospective cohort study with 289 pregnant women recruited between weeks 10–12 of gestation in a hospital of Granada, Spain, and associations between 25-hydroxyvitamin D, PTH, calcium, phosphorus, and perinatal adverse outcomes, namely preterm birth, low birth weight and small for gestational age were evaluated. We found a trend towards lower maternal 25-hydroxyvitamin D serum levels in the first trimester of gestation and higher odds of preterm birth. This association was stronger amongst women with elevated levels of PTH (>80th percentile), and it was not attenuated after adjusting for preterm birth confounders. Although a similar association was observed for low birth weight, it was not statistically significant after confounder adjustment. SGA was defined based on weight and weeks of gestation at delivery from Spanish percentile charts [39] did not correlate either with vitamin D or related metabolites.

With the exemption of low birth weight, the prevalence of pregnancy and perinatal adverse outcomes was lower than average estimates in Europe. Preeclampsia is a strong contributor to preterm birth [49]. The small number of preeclampsia cases could partially explain the low preterm birth cases observed in the cohort of study.

### 4.1. Limitations of the Study

The present study has some limitations to be acknowledged. The prevalence of the main outcome of the study, preterm birth, was more than 30% lower than average estimates in Europe. This could be the cause of the lack of significance observed in the association between vitamin D deficiency and preterm birth and might compromise extrapolation of our results to other populations. Regarding secondary outcomes (SGA and LBW), it is possible that the lack of statistical significance of associations could be consequence of sample size limitations given that they were not included in sample size calculations. We did not use liquid chromatography-tandem mass spectrometry (LC-MS/MS), which is considered the gold-standard method by most authors to analyze 25-hydroxyvitamin D. Due to equipment limitations, we did not directly measure ionized calcium and we could not determine albumin levels thus we were not able to estimate ionized calcium concentration which is the most active form of calcium. Additionally, we could not measure other important bone turnover biomarkers such as alkaline phosphatase which would be of interest when assessing associations between 25-hydroxyvitamin D and PTH. Almost 75% of the samples were obtained during autumn, and all participants were Caucasian. Therefore, it was not possible to adjust the results for ethnicity, and seasonality adjustment could be inaccurate. These are important factors that can potentially influence maternal vitamin D blood levels [22].

### 4.2. Deficiency of Vitamin D and Preterm Birth

Spain is a Mediterranean country with high levels of sun exposure. Despite this fact, vitamin D deficiency is highly prevalent among Spanish pregnant women [50]. This situation is known as the “Mediterranean paradox,” and it has been estimated that 41%–90% of all pregnant women living in Mediterranean countries have vitamin D levels below sufficiency [51]. In line with this data, only 11.76% of study participants had sufficient vitamin D levels (>30 ng/mL) whilst more than one-third of the women had levels below 20 ng/mL, which implies a high prevalence of vitamin D deficiency amongst participants. The observed ratio of vitamin D sufficiency/insufficiency is consistent with the results obtained by Perez-Ferre et al., who conducted a prospective cohort study in 266 pregnant women during weeks 24–28 of gestation in Madrid, Spain, finding a significant association between vitamin D deficiency and preterm birth using the same vitamin D cut-off points, in both unadjusted and adjusted logistic regression models (OR = 3.31, 95% CI (1.52, 7.19), *p* = 0.002/aOR = 3.80, 95% CI (1.32, 10.97), *p* = 0.013) [31]. However, in the present study, we only observed a statistically significant association between vitamin D deficiency (<20 ng/mL) and preterm birth after adjusting for confounders with statistical significance in the univariate model (*p* < 0.20). Differences between both studies could be attributed to our significantly smaller number of PTB cases and different sampling time. Another study conducted in a Spanish cohort of 2382 pregnant women could not find any association between 25-hydroxyvitamin D and perinatal outcomes, including PTB and SGA. However, almost 50% of the participants had sufficient levels of vitamin D, which implies a low rate of vitamin D insufficiency in comparison with average estimates [52].

Using similar study designs, several authors have explored the link between vitamin D deficiency and prematurity in other countries yielding negative results [10,53], whilst other studies have found a positive association [11,54]. Authors of these studies state the necessity of conducting well-designed randomized clinical trials to further clarify this subject. However, meta-analyses of randomized clinical trials have failed to verify an association between vitamin D supplementation and lower odds of preterm birth [4,55]. In this sense, randomized clinical trials conducted to date not only have to face important ethical issues but also lack relevant criteria related to nutrients studies [56]. One important criterium that is usually overlooked is the optimization of the status of associated nutrients in order to ensure the causality of observed associations [57].

### 4.3. Vitamin D Associated Metabolites and Perinatal Outcomes

Vitamin D regulates calcium and phosphorus homeostasis, and its production is controlled by PTH [58]. Santorelli et al. measured 25-hydroxyvitamin D, PTH, and calcium in a heterogeneous population composed of 1010 pregnant women differentiating between white and Pakistani participants. They observed that higher calcium levels were associated with lower odds of PTB amongst white participants, whilst vitamin D exerted a protective effect on the overall risk of SGA. However, none of the studied metabolites were associated with SGA in white participants [17]. In the present study, we did not observe any significant association between calcium and preterm birth in Caucasian pregnant women. Nonetheless, due to sample limitations, we were not able to examine the impact that ethnicity could have on the analyses.

Other authors have explored the concept of functional vitamin D deficiency in pregnancy as a cause of calcium metabolic stress, which could ultimately lead to perinatal adverse outcomes associated with the deficiency of this secosteroid. This concept has been applied to examine the association between vitamin D deficiency, gestational hypertensive disorders, and fetal growth restriction [26,29]. Scholl et al. observed a higher incidence of SGA cases amongst pregnant women with PTH > 62 pg/mL in combination with 25-hydroxyvitamin D < 20 ng/mL or calcium intakes below 60% of the estimated average requirement (OR = 2.23, 95% CI (1.23, 4.33)) [29]. In the same line, Hemmingway et al. found a 2.38-fold increased risk of SGA amongst pregnant women with serum 25-hydroxyvitamin D levels < 12 ng/mL (<30 nmol/L) in combination with PTH > 80th percentile in the cohort of study (RR = 2.38, 95% CI (1.31, 4.33)). However, this association was not statistically significant after confounder adjustment [26]. More recently, Meng et al. prospectively measured PTH, calcium, and 25-hydroxyvitamin D in 3407 participants in China, finding that maternal 25-hydroxyvitamin D levels <12 ng/mL and <20 ng/mL (<50 nmol/L) along with PTH concentrations >75th percentile were associated with increased risk of SGA and lower mean birth weight compared to vitamin D sufficient women. This association was not attenuated in sensitivity analyses (PTH > 80th percentile) [25]. On the other hand, Tao et al. evaluated the effect of the duration of vitamin D supplementation (400–600 IU/d) on fetal growth, finding a direct association between more prolonged vitamin D supplementation and higher weeks of gestation and weight at delivery independently of calcium and phosphorus concentrations [59]. In the present study, we found a correlation between low birth weight and vitamin D < 20 ng/mL in combination with high levels of PTH (>80th). However, this association was not significant after adjustment for confounders, which implies that gestational age at delivery was the main underlying factor for the association. In the same fashion, the risk of SGA was not correlated with vitamin D or PTH in any subgroup analysis. Nonetheless, we observed that women with PTH levels > 80th percentile and 25-hydroxyvitamin D < 20 ng/mL had more than five times higher odds of PTB compared to the reference group, and this relationship persisted after adjusting for confounders. These results were consistent with those obtained in the sensitivity analysis using the 75th PTH percentile instead Appendix A. It is possible that vitamin D deficiency could exert an effect on birth weight by influencing the length of gestation [23]. Finally, neither calcium nor phosphorus concentrations were associated with any studied outcome.

Our results do not support the hypothesis that elevated levels of PTH in combination with vitamin D deficiency are associated with fetal growth restriction. However, reference levels for PTH during pregnancy are not firmly established, and SGA is defined depending on specific reference charts and, thus, results could not be extrapolated to other populations.

## 5. Conclusions

In the present study, we observed that vitamin D deficiency defined as 25-hydroxyvitamin D concentrations below 20 ng/mL, in combination with parathyroid hormone maternal levels above the 80th percentile during the first trimester of gestation, was a better estimator of preterm birth than the assessment of vitamin D deficiency in isolation. However, we did not observe the same association with low birth weight after controlling for weeks of gestation or small for gestational age. Interventional studies with vitamin D supplementation would benefit from measuring parathyroid hormone in order to demonstrate a potential causal association between deficiency of vitamin D and perinatal adverse outcomes.

## Figures and Tables

**Figure 1 nutrients-12-03279-f001:**
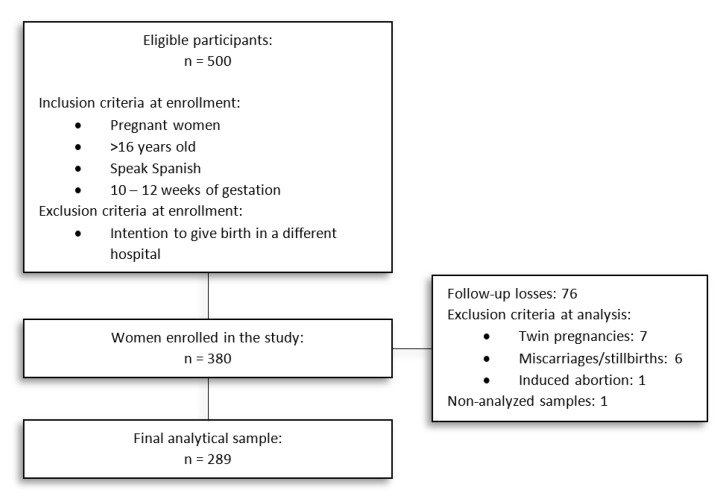
Flow diagram of participants.

**Figure 2 nutrients-12-03279-f002:**
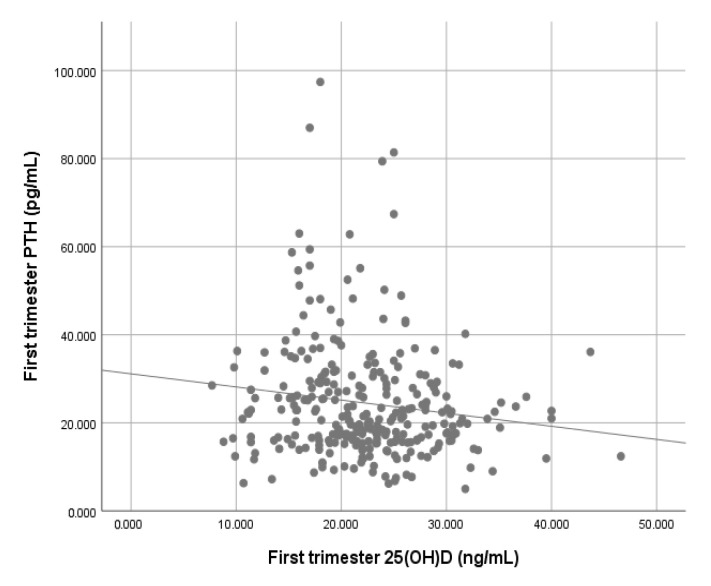
Scatter plot of the correlation between parathyroid hormone and vitamin D. PTH: parathyroid hormone; 25(OH)D: 25-hydroxyvtiamin D.

**Table 1 nutrients-12-03279-t001:** Characteristics of participants based on 25-hydroxyvitamin D cut-off points.

Variable	All Participants (*n* = 289)	Vitamin D < 20 ng/mL (*n* = 105)	Vitamin D ≥ 20 ng/mL (*n* = 184)	*p*-Value
**Age**	33 (29–36)	34 (30–35.5)	32 (28–36)	0.358
**Seasonality**				0.159
Spring	42 (14.5%)	12 (11.4%)	30 (16.3%)	
Summer	21 (7.3%)	5 (4.8%)	16 (8.7%)	
Autumn	216 (74.7%)	82 (78.1%)	134 (72.83%)	
Winter	10 (3.5%)	6 (5.7%)	4 (2.17%)	
**Smoking**				0.976
Yes	36 (12.5%)	13 (12.4%)	23 (12.5%)	
No	253 (87.5%)	92 (87.6%)	161 (87.5%)	
**Obesity**				0.01 *
BMI ≥ 30	52 (18%)	27 (25.7%)	25 (13.59%)	
BMI < 30	237 (82%)	78 (74.3%)	159 (86.41%)	
**Parity**				0.198
Nulliparity	163 (56.4%)	54 (51.4%)	75 (40.76%)	
Multiparity	126 (43.6%)	51 (48.6%)	109 (59.24%)	
**Preterm birth**	17 (5.9%)	10 (9.5%)	7(3.8%)	0.047 *
**Low birth weight**	24 (8.3%)	13 (12.4%)	11 (6%)	0.058
**Small for gestational age**	27 (9.3%)	14 (13.3%)	13 (7.1%)	0.078

BMI: body mass index. Categorical data are presented as absolute frequency (percentages), and continuous data are presented as median (interquartile range). *p*-values were obtained by Pearson’s χ^2^ test for categorical variables and the Mann–Whitney U test for continuous variables. * *p*-value < 0.05.

**Table 2 nutrients-12-03279-t002:** Concentrations of parathyroid hormone, calcium and phosphorus based on 25-hydroxyvitamin D cut-off points.

Metabolite	All Participants (*n* = 289)	Vitamin D < 20 ng/mL (*n* = 105)	Vitamin D ≥ 20 ng/mL (*n* = 184)	*p*-Value
**Parathyroid hormone**	21 (16–29.3) pg/mL	25.50 (16.6–34.6) pg/mL	19.6 (15.8–26.4) pg/mL	0.002 *
**Calcium**	9.2 (8.9–9.4) mg/dL	9.2 (9–9.4) mg/dL	9.2 (8.9–9.4) mg/dL	0.914
**Phosphorus**	3.6 (3.4–3.9) mg/dL	3.6 (3.4–3.9) mg/dL	3.6 (3.4–3.9) mg/dL	0.899

Data are presented as median (interquartile range). *p*-values were obtained using the Mann–Whitney U test for continuous variables. * *p*-value < 0.05.

**Table 3 nutrients-12-03279-t003:** Frequency of pregnancy and perinatal adverse outcomes compared to global and regional frequencies.

Outcome	Frequency	Estimated Global Frequency	Estimated Frequency in the USA	Estimated Frequency in Europe
**Preeclampsia**	1.7%	4.6% (2010) [42]	3% (2010) [42]	5.3% (2010) [42]
**Gestational diabetes mellitus**	5.6%	16.9% (2013) [43]	4.6–9.2% (2010) [44]	15.2% (2013) [43]
**Cesarean section**	21.5%	31% (2011) [45]	31.9% (2018) [46]	25.2% (2010) [47]
**Preterm birth**	5.9%	10.6% (2014) [5]	10.2% (2018) [46]	8.7% (2014) [5]
**Low birthweight**	8.3%	14.6% (2015) [48]	8.28% (2018) [46]	6.5% (2015) [48]

**Table 4 nutrients-12-03279-t004:** Associations between Vitamin D deficiency, parathyroid hormone (PTH), calcium, phosphorus, and adverse perinatal outcomes in the cohort of study.

Outcome		Vitamin D Deficiency	*p*-Value	Parathyroid Hormone	*p*-Value	Calcium	*p*-Value	Phosphorus	*p*-Value
**Preterm Birth**	Unadjusted OR	2.662 (0.982–7.217)	0.054	1.030 (1.002–1.058)	0.035 *	2.024 (0.581–7.048)	0.268	1.021 (0.630–1.652)	0.934
	Adjusted OR ^1^	3.529 (1.159–10.741)	0.026 *	1.027 (0.997–1.059)	0.083	1.814 (0.513–6.413)	0.355	0.764 (0.240–2.431)	0.648
**Low Birth Weight**	Unadjusted OR	2.222 (0.958–5.157)	0.063	1.019 (0.993–1.046)	0.156	1.572 (0.566–4.366)	0.386	0.738 (0.282–1.927)	0.535
	Adjusted OR ^2^	1.586 (0.586–4.336)	0.361	1.009 (0.977–1.041)	0.597	1.212 (0.355–4.144)	0.758	0.568 (0.189–1.711)	0.315
**Small for Gestational Age**	Unadjusted OR	2.024 (0.912–4.488)	0.083	0.985 (0.951–1.020)	0.399	1.215 (0.488–3.022)	0.676	0.735 (0.296–1.913)	0.551
	Adjusted OR ^3^	1.794 (0.786–4.093)	0.165	0.978 (0.939–1.018)	0.276	1.127 (0.435–2.923)	0.805	0.699 (0.269–1.818)	0.463

Data reported as odds ratios (OR) (95%CI). ^1^ Adjusted for: history of PTB and pre-eclampsia. ^2^ Adjusted for: maternal age, smoking habit, pre-eclampsia, and preterm birth. ^3^ Adjusted for: seasonality, smoking habit, and parity. * *p*-value < 0.05.

**Table 5 nutrients-12-03279-t005:** Associations between combination of maternal serum 25-hydroxyvitamin D and PTH 80th percentile and perinatal adverse outcomes.

	Preterm Birth			Low Birth Weight			Small for GESTATIONAL Age		
	*n* (%)	OR	aOR ^1^	*n* (%)	OR	aOR ^2^	*n* (%)	OR	aOR ^3^
**25[OH]D ≥ 20 ng/mL (≥50 nmol/L)**									
**PTH > 80th**	0/26 (0%)	--	--	0/26 0%	--	--	0/26 0%	--	--
**PTH ≤ 80th**	7/158 (4.4%)	0.561 (0.207–1.517)	0.581 (0.203–1.667)	11/158 (7%)	0.679 (0.294–1.571)	0.899 (0.333–2.432)	13/158 (8.2%)	0.749 (0.339–1.656)	0.857 (0.376–1.954)
**25[OH]D < 20 ng/mL (<50 nmol/L)**									
**PTH > 80th**	6/31 (19.4%)	5.389 (1.837–15.812) *	6.223 (1.939–19.970) *	7/31 (19.4%)	4.135 (1.560–10.963) *	2.653 (0.766–9.188)	4/31 (12.9%)	1.514 (0.487–4.705)	1.356 [0.407–4.518]
**PTH ≤ 80th**	4/74 (5.4%)	0.888 (0.280–2.813)	1.057 (0.313–1.357)	6/74 (8.1%)	0.966 (0.368–2.533)	0.877 (0.268–2.868)	10/74 (13.5%)	1.820 (0.793–4.175)	1.663 (0.705–3.919)
**25[OH]D ≥ 30 ng/mL (≥75 nmol/L)**									
**PTH > 80th**	0/4 (0%)	--	--	0/4 0%	--	--	0/4 0%	--	--
**PTH ≤ 80th**	2/30 (6.7%)	1.162 (0.253–5.346)	1.480 (0.310–7.065)	1/30 (3.3%)	0.354 (0.460–2.718)	0.257 (0.024–2.787)	1/30 (3.3%)	0.309 (0.040–2.363)	0.324 (0.041–2.548)
**25[OH]D < 30 ng/mL (<75 nmol/L)**									
**PTH > 80th**	6/53 (11.3%)	2.611 (0.920–7.411)	2.109 (0.673–6.611)	7/53 (13.2%)	1.960 (0.769–4.998)	1.402 (0.442–4.441)	4/53 (7.5%)	0.756 (0.250–2.285)	0.713 (0.226–2.251)
**PTH ≤ 80th**	9/202 (4.5%)	0.460 (0.172–1.236)	0.492 (0.171–1.419)	16/202 (7.9%)	0.849 (0.349–2.066)	1.188 (0.391–3.615)	22/202 (10.9%)	2.004 (0.733–5.479)	2.202 (0.772–6.181)

Data reported as OR (95%CI). OR: odds ratios. aOR: adjusted odds ratio. ^1^ Adjusted for pre-eclampsia and history of preterm birth. ^2^ Adjusted for maternal age, smoking habit, pre-eclampsia, and preterm birth. ^3^ Adjusted for seasonality, smoking habit, and parity. * *p*-value < 0.05.

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
