# Peer review of "Associations of Vitamin D Deficiency, Parathyroid hormone, Calcium, and Phosphorus with Perinatal Adverse Outcomes. A Prospective Cohort Study"

_nutrients, 2020, doi:10.3390/nu12113279_

Round 1
Reviewer 1 Report
This is a written paper exploring the association between vitamin D and parathyroid hormone status with perinatal outcomes. The generalizability is limited as this was conducted with a sample of 289 pregnant women attending one hospital center in Granada, Spain (100% Caucasian). However, it does provide support for the combined use of parathyroid hormone with 25-hydrxyvitamin D as an estimator of preterm birth. The following concerns however should be addressed. Please correct small grammatical errors, i.e. parathormone instead or parathyroid hormone.
Introduction: Although, the rationale for the use of parathyroid hormone is clear, the authors have not provided a justification for the selection of calcium and phosphorus as metabolites of interest. And why were these measures selected as compared to alkaline phosphatase or ionized calcium which have typically been used to assess for vitamin D status in bone disease and also found to be associated with other health risks.
Methods: Please confirm whether this was a secondary analysis from another study with a different aim.
Sample size: Justify the sample size given that the incidence of preterm birth in the sample was lower than anticipated, ~5.9% (vs. lines 87-94). Also, were other outcomes (SGA, LBW) included in the sample size estimation i.e. were you sufficiently powered to test these other outcomes or could this explain the lack of association found?
Exposure Data (lines 102-126):
The term “vitamin D” has been used incorrectly in this section and please check throughout the paper. I believe you are referring to 25-hydroxyvitamin D (calcidiol) and not vitamin D (cholecalciferol). Also, the classifications used for vitamin D status are incorrect. The IOM defines vitamin D deficiency as 25(OH)D concentrations <30 nmol/L (12 ng/mL), inadequacy as 30-50 nmol/L (12–20 ng/mL) and sufficiency as 50 nmol/L (20 ng/mL) or more. Just because other studies are using these definitions does not make it correct.
Also, the authors have opted to use the 80% percentile as elevated PTH. Please justify why the deflection point of the PTH-25(OH)D regression line was not used instead, as this may have been a better estimate of the optimal threshold.
Statistical Analysis: This section does not make reference to how calcium and phosphate were analyzed in logistic regression models. Please also clarify the dependent variable (column titles) in Table 3 i.e. was parathyroid hormone a categorical or continuous variable?
Table 1: Can you add the % PTB, LBW and SGA to this table. It would be interesting to have this data presented by vitamin D group. Also, was supplementation considered at baseline including high dose vitamin D.
Reviewer 2 Report
Perez-Castillo et al. study aimed to examine the association between vitamin D deficiency, parathormone, calcium, and phosphorus during early pregnancy with preterm birth, low birth weight, and small for gestational age in Spanish women.
This is an original study that has a great clinical impact and the ability to change pregnant women behaviours. Vitamin D deficiency in pregnancy is a major problem all the World and can be linked to perinatal adverse outcomes.
I appreciate the Authors' contribution to this research and the interesting results obtained.
However, there are several areas that need to be addressed to help improve the paper.
Please find below general comments:
What is the aim of the study?
Is the secondary outcome of the study “to evaluate the role that metabolites related to the metabolism of this secosteroid, namely PTH, calcium, and phosphorus could play in the associations"?
I propose to clarify the aim of the study and harmonize it with the abstract.
I suggest that authors should include a hypothesis of the study.
The Materials and Methods section should be improved.
lines 74 and 82: It should be specified whether this study (women were recruited during 2018-2019) was part of a prospective cohort study conducted from 2017 to 2020? It is not clear now.
Section 2.2. Exposure Data: Please think about the title - maybe better: Clinical and biochemical procedures/analysis.
I also propose to explain how the data of maternal diastolic and systolic blood pressure, proteinuria, and glucose tolerance test was collected and move it to section 2.2
Results:
Section 3.1; lines 160-167, should be in the Materials and Methods and linked to the Participants data section, lines 82-86.
I would also suggest redrawing: Flow diagram of participants, including the inclusion/exclusion criteria for the study.
Lines 179-182 – they are inclusion/exclusion criteria, please move them to the Participants data section.
Table 1: Please explain if all women were obese?
Moreover, data of parathormone, calcium, and phosphorus should be separated (otherwise written).
In Table 2, the Cases column should be removed and this table might be better to include it in the Discussion section.
Table 4 is not easy to follow. Please reformulate the table to improve readability.
The discussion should be carefully read and checked again.
Round 2
Reviewer 1 Report
Thank you for addressing my comments and no further changes are requested.
Reviewer 2 Report
Dear Authors,
Thank you for considering my comments and making the appropriate changes in the manuscript.
The revised manuscript is substantially improved.
I can now support its application.
Best regards,